# Orbital- and millennial-scale Asian winter monsoon variability across the Pliocene–Pleistocene glacial intensification

Hong Ao [1,2] ✉, Diederik Liebrand[3], Mark J. Dekkers[4], Andrew P. Roberts [5], Tara N. Jonell [6], Zhangdong Jin [1,2], Yougui Song [1], Qingsong Liu[7], Qiang Sun[8], Xinxia Li[1,9], Chunju Huang [9], Xiaoke Qiang [1] & Peng Zhang [1,2]

Intensification of northern hemisphere glaciation (iNHG), ~2.7 million years ago (Ma), led to establishment of the Pleistocene to present-day bipolar ice-house state. Here we document evolution of orbital- and millennial-scale Asian winter monsoon (AWM) variability across the iNHG using a palaeomagnetically dated centennial-resolution grain size record between 3.6 and 1.9 Ma from a previously undescribed loess-palaeosol/red clay section on the central Chinese Loess Plateau. We find that the late Pliocene–early Pleistocene AWM was characterized by combined 41-kyr and ~100-kyr cycles, in response to ice volume and atmospheric $CO_2$ forcing. Northern hemisphere ice sheet expansion, which was accompanied by an atmospheric $CO_2$ concentration decline, substantially increased glacial AWM intensity and its orbitally oscillating amplitudes across the iNHG. Superposed on orbital variability, we find that millennial AWM intensity fluctuations persisted during both the warmer (higher-$CO_2$) late Pliocene and colder (lower-$CO_2$) early Pleistocene, in response to both external astronomical forcing and internal climate dynamics.

Sustained anthropogenic carbon emissions are causing Earth to mimic a warm climate state most recently experienced in the late Pliocene (Piacenzian stage) 3.6–2.58 million years ago (Ma)[1]. In particular, during the mid-Piacenzian warm period at ~3.3–3 Ma, globally averaged temperatures were 2–4 °C higher than preindustrial conditions[2–4] and global mean sea levels were ~20 m above the present level[5,6], yet atmospheric $CO_2$ concentrations were comparable to present-day values[1,3,4,7–10]. Following this climatic optimum, global climate cooled and high-latitude northern hemisphere (NH) regions became increasingly glaciated from ~3 Ma onward[1]. Notably, distinct and widespread intensification of NH glaciation (iNHG) occurred across the Pliocene–Pleistocene transition (2.58 Ma), which was marked by

development of large, thick ice sheets in Greenland, North America, and Eurasia, while ice-rafted debris inputs to the North Pacific and North Atlantic Oceans increased substantially[1,5,6]. The iNHG marks a threshold in the long-term cooling trend to a well-established bipolar icehouse state that persists today[1,5,11]. High-resolution climate reconstructions across the iNHG are crucial for better understanding past, modern, and future climate processes under a broader range of climate-cryosphere boundary conditions beyond the permanent bipolar icehouse state that marked the Pleistocene. However, orbital- and millennial-scale climate variability remains poorly constrained across the iNHG, especially within continental interior settings that host much of the present global population. Millennial variability in

[1]State Key Laboratory of Loess and Quaternary Geology, Institute of Earth Environment, Chinese Academy of Sciences, Xi'an, China. [2]Laoshan Laboratory, Qingdao, China. [3]Department of Earth and Environmental Sciences, University of Manchester, Manchester, UK. [4]Paleomagnetic Laboratory 'Fort Hoofddijk', Department of Earth Sciences, Faculty of Geosciences, Utrecht University, Utrecht, The Netherlands. [5]Research School of Earth Sciences, Australian National University, Canberra, ACT, Australia. [6]School of Geographical and Earth Sciences, University of Glasgow, Glasgow, UK. [7]Department of Ocean Science and Engineering, Southern University of Science and Technology, Shenzhen, China. [8]College of Geology and Environment, Xi'an University of Science and Technology, Xi'an, China. [9]School of Earth Sciences, China University of Geosciences (Wuhan), Wuhan, China. ✉e-mail: aohong@ieecas.cn

mean annual air temperature oscillations throughout the last glacial cycle is reconstructed from Greenland ice cores (i.e., Dansgaard–Oeschger cycles)[12] and from multiple centennial-resolution records from the North Atlantic Ocean, Mediterranean, Iberian margin, Balkan Peninsula (Lake Ohrid), and Chinese Loess Plateau (CLP) throughout the last 1.5 million years (Myr)[13–16]. However, millennial climate variability before 1.5 Ma remains underexplored in both oceanic and terrestrial settings.

As the largest monsoon system on Earth, the Asian monsoon determines climatic and environmental conditions over the South–East Asian continent, large parts of which are populated densely[17]. During boreal winters, cold air from the Siberian High pressure cell over the mid- to high-latitude Asian continental interior induces northwesterly Asian winter monsoon (AWM) advection (Fig. 1a). Conversely, the Asian summer monsoon (ASM) transports heat and moisture from the western Pacific and Indian Oceans across South Asia and tropical East Asia to North China and Japan during summers (Supplementary Fig. 1a).

The CLP is located at the transition between humid and arid regions in north-central China and harbors a vast expanse (~640,000 km²) of thick aeolian dust deposits (up to 1 km thickness). These deposits comprise a Neogene red clay sequence stratigraphically succeeded by a Quaternary sequence of alternating yellow loess and red palaeosol layers[18–21]. Magneto-stratigraphic results suggest that the transition from red clay to loess-palaeosol sequences across the CLP occurred synchronously around the Gauss–Matuyama boundary at 2.58 Ma[20–26], which enables robustly-dated studies of climate changes across the iNHG. These

dust deposits are sourced from the inland Gobi Desert and other nearby sandy deserts and poorly-vegetated areas and transported primarily by the near-surface northwesterly AWM[27] (Fig. 1b). Grain-size variability of these wind-blown dust deposits is largely controlled by AWM intensity, with additional influence of transport distance and changes in sediment source regions[21,23,28–31]. High-resolution grain size records spanning from the Pliocene red clay to Pleistocene loess-palaeosol sequences can, therefore, provide indications of detailed temporal AWM intensity variations across the iNHG. CLP grain size studies suggest distinct orbital-scale Pleistocene AWM variability, persistent millennial-scale variability over the last 1.5 Myr, and stepwise mean state intensifications across the iNHG and mid-Pleistocene transition (MPT, 1.25–0.6 Ma)[13,19,23,30,32–37]. However, these studies lack detailed assessment of concurrent orbital- and millennial-scale AWM variability across the iNHG. To fill this gap, here we present a centennial-resolution CLP grain size record, which is dated between 3.6 and 1.9 Ma by magnetostratigraphy, to assess (1) secular evolution, (2) orbital- and millennial-scale variability, and (3) the underlying AWM dynamics across the iNHG that is marked by substantial climate-cryosphere boundary condition changes. By doing so, we extend the oldest record of co-occurring orbital- and millennial-scale AWM variability[13] by >2 Myr, into the relatively warmer Pliocene.

## Results and discussion
### Centennial-resolution late Pliocene–early Pleistocene AWM reconstruction

Our study site, the Chongxin section (35°15′N, 107°8′E), is located on the central CLP under the influence of seasonally alternating AWM and

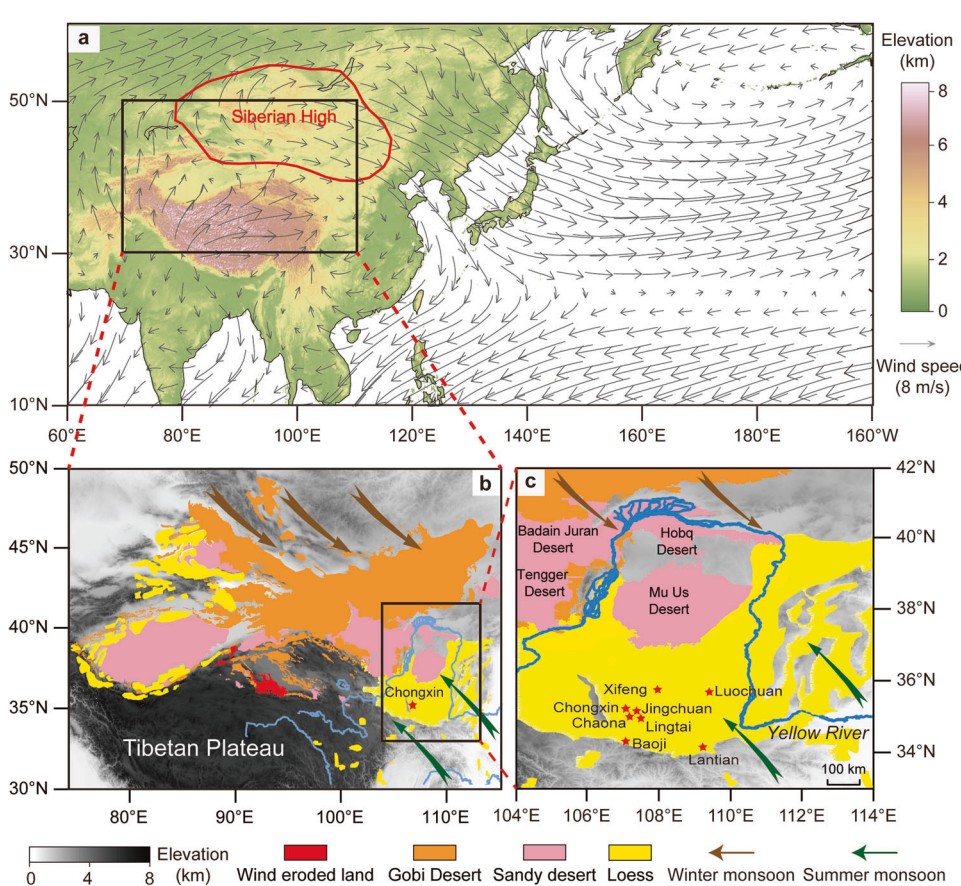

**Fig. 1 | Study site location and boreal winter atmospheric circulation. a** Asian topographic map with the centre of the Siberian High pressure cell (mean sea level pressure exceeding 1028 hPa, red circle) and boreal winter monsoon winds (925 hPa, grey arrows) based on the National Center for Environmental Prediction/Department of Energy (NCEP/DOE) Reanalysis 2 (NCEP R2) between 1979 and 2020. **b** Topographic map of Asian dust sources (Tibetan Plateau, Gobi Desert, sandy deserts, and wind-eroded land) and aeolian loess deposits in North China[27]. **c** Topographic map of the Chinese Loess Plateau. Red stars represent red clay/loess-palaeosol sections mentioned in the text. We created these maps with ArcGIS (version 10.7) and Adobe Illustrator 2020 software.

ASM circulation (Fig. 1c). The central CLP is covered with quasi-continuous Pleistocene loess-palaeosol and Pliocene red clay sequences[20,21,32] compared to some sections on the northern (desert) margin of the CLP which may feature erosional hiatuses[38]. In the Pleistocene loess-palaeosol sequence, loess layers that formed during colder/drier intervals with lower ASM precipitation and stronger AWM winds are yellow, whereas intercalated palaeosol layers that formed during warmer/wetter intervals with higher ASM precipitation and weaker AWM winds are light red (Supplementary Fig. 1b). The underlying red clay, which was deposited under warmer and more sustained moist Pliocene conditions, has a higher saturated red colouration than the Pleistocene palaeosols (Supplementary Fig. 1b). We collected 3571 unoriented samples from the Chongxin loess-palaeosol/red clay section, which spans from the late Pliocene to early Pleistocene, at 2 cm intervals for grain size analysis. This large dataset resolves grain size variations on an average temporal resolution of ~0.5 kyr, which is unprecedented in both marine and terrestrial realms for this time interval. It enables detailed assessment of orbital- and millennial-scale AWM dynamics across the iNHG. We also collected 251 oriented block samples for magnetostratigraphic analysis.

Detailed rock magnetic and palaeomagnetic analyses establish a magnetochronology for the Chongxin section ("Methods"; Fig. 2a and Supplementary Figs. 2–6). Magnetite was identified as the major characteristic remanent magnetization (ChRM) carrier ("Methods"; Supplementary Figs. 2–6). Following removal of a low-temperature secondary overprint at temperatures up to 200–300 °C, a ChRM was isolated during subsequent stepwise thermal demagnetization up to 620 °C (Supplementary Fig. 6). From 251 demagnetized samples, 212 yielded stable ChRM directions, from which virtual geomagnetic pole (VGP) latitudes were calculated to establish a magnetostratigraphic zonation. The Chongxin magnetostratigraphic sequence has a distinct pattern of five normal and five reverse polarity zones, spanning from just below the Gilbert–Gauss reversal boundary to the normal polarity Olduvai subchron of the geomagnetic polarity timescale (GPTS)[39] (Fig. 2a, b and Supplementary Fig. 7a, b). This magnetochronology is consistent with magnetostratigraphic correlations of other CLP red clay/loess-palaeosol sections[20,21,23–26] (Supplementary Fig. 7). The Gauss–Matuyama boundary is located consistently around the transition from red clay to loess-palaeosol sequences. The base of the Olduvai subchron is situated (1) around the $S_{26}$–$L_{26}$ boundary in the Chongxin section, (2) in the lower part of palaeosol $S_{26}$ in the Jingchuan[20], Xifeng[23], and Lingtai sections[21], and (3) in the upper part of loess $L_{27}$ in the Baoji section[20]. This slight displacement of geomagnetic polarity reversals in different sections may be linked to slightly variable post-depositional remanent magnetization lock-in depth across the CLP[40] and/or different sampling intervals of magnetostratigraphic records. The short-duration Réunion geomagnetic excursion is not registered in most studied loess-palaeosol sections[20,24], but appears clearly in the Chongxin section (Fig. 2a and Supplementary Fig. 7). We identify the Réunion excursion between the upper part of loess $L_{28}$ and the lower part of palaeosol $S_{27}$, broadly consistent with its location within loess $L_{28}$ in the Shangchen section on the southern CLP margin[41]. The early Pleistocene loess-palaeosol sequence has a higher sedimentation rate than the underlying red clay and corresponds to the long Matuyama reverse polarity chron between the Gauss–Matuyama reversal boundary and the normal polarity Olduvai subchron. Consequently, the early Pleistocene loess-palaeosol sequence (in the Matuyama polarity chron) has a relatively coarser palaeomagnetic sample spacing in the depth domain, compared to the Pliocene red clay sequence in the Gauss polarity chron. However, our Matuyama polarity chron data are sufficiently dense to enable straightforward correlation to the GPTS and other CLP red clay/loess-palaeosol magnetostratigraphic sequences (Fig. 2a, b and Supplementary Fig. 7). Based on correlation to the 2020 GPTS[39], linear interpolation between subsequent tie points using the identified

geomagnetic polarity reversals yields a magnetochronology between ~3.60 and 1.94 Ma, i.e., a late Pliocene to early Pleistocene time interval (Piacenzian and Gelasian stages). Astronomical tuning is a routine approach for constructing more precise and accurate age models for CLP loess-palaeosol/red clay sequences, especially once an initial magnetostratigraphic age model has been developed[20,23,42]. We refined the Chongxin magnetostratigraphic age model by obliquity-tuning of the bulk sample mean grain size (MGS) record (AWM indicator) using an automatic orbital tuning procedure[43] ("Methods"; Supplementary Figs. 8 and 9).

MGS records from bulk samples and extracted quartz particles have almost identical orbital cycles for both Pliocene red clay and Pleistocene loess-palaeosol sequences (see Fig. 2 in ref. 21), albeit with some differences in the magnitude of glacial–interglacial pacing and interglacial minima[21,23,36]. Overall consistency of orbital cycles suggests that the MGS of CLP bulk samples is as effective as MGS from extracted quartz particles in recording AWM intensity, without significant pedogenic overprinting. We, therefore, primarily employ the more commonly used bulk red clay and loess-palaeosol grain size[13,19,20,30,31,35] to assess AWM variations. We measured grain size distributions for all 3571 unoriented bulk samples after laboratory organic matter and carbonate removal ("Methods").

The Chongxin MGS record has distinct glacial-interglacial variability throughout the 3.6–1.9 Ma interval, which is broadly comparable to global mean sea level reconstructed from deep-sea-carbonate-microfossil–based $\delta^{18}O$ records[5] (Fig. 2c, e). Subtle differences are also observed in exact glacial-interglacial cycle shapes, which are, for example, more symmetrical in the sea level record. Generally, glacials have coarser grain sizes and stronger AWM winds than interglacials (Fig. 2c), consistent with Pleistocene observations and theory/model climate predictions[13,21,30,31,44]. Glacial MGS values increase markedly during marine isotope stage (MIS) G5 around 2.7 Ma and become even coarser after 2.6 Ma (Fig. 2c), which is indicative of increased dust transport capacity by stronger AWM winds from drier source regions to the north and west of the CLP under colder conditions across the iNHG[21,30,31,36,45]. Such glacial loess coarsening induced larger orbital-scale MGS oscillations across the iNHG (Fig. 2c). Generally, the early Pleistocene loess-palaeosol sequence has slightly coarser interglacial MGS values (~7–11 μm) but substantially coarser glacial MGS values (~11–14 μm) than the Pliocene red clay (typically ~7–8 μm in interglacials and ~8–10 μm in glacials), which is consistent with greater AWM intensification during glacials compared to interglacials across the iNHG. The larger orbital-scale Pleistocene MGS oscillations are more distinct in the <200 kyr filtered MGS record (Fig. 2d; "Methods"). Similarly increased glacial-interglacial amplitudes across the iNHG are also observed in the global mean sea level record[5] and its <200 kyr filtered counterpart (Fig. 2e, f; "Methods"). Cross-wavelet and cross-power spectral analyses of the Chongxin MGS and global mean sea level[5] records suggest that they have high coherency on the ~100-kyr and 41-kyr bands (Fig. 3), although their phase relationship varies over time (Fig. 3a, arrow directions vary through time). Cross-wavelet spectral analysis further confirms that the 41-kyr and ~100-kyr signals are enhanced across the iNHG, consistent with increased orbitally oscillating amplitudes (Figs. 2 and 3). Cross-power spectral analysis suggests that the 41-kyr band is statistically higher than the ~100-kyr band.

The Pliocene red clay sequence has thicker more homogeneous palaeosols that are redder in color and lack thick yellow loess intercalations typical of the overlying Pleistocene loess-palaeosol sequence (Supplementary Fig. 1b). Consistent with MGS changes across the iNHG, the red clay sequence transitioned to the loess-palaeosol sequence synchronously at Gauss–Matuyama boundary time as is documented across the entire CLP (Supplementary Fig. 7). The rapid lithological shift across the iNHG is a consequence of both glacial AWM strengthening (Fig. 2c) that increased dust transport and accumulation

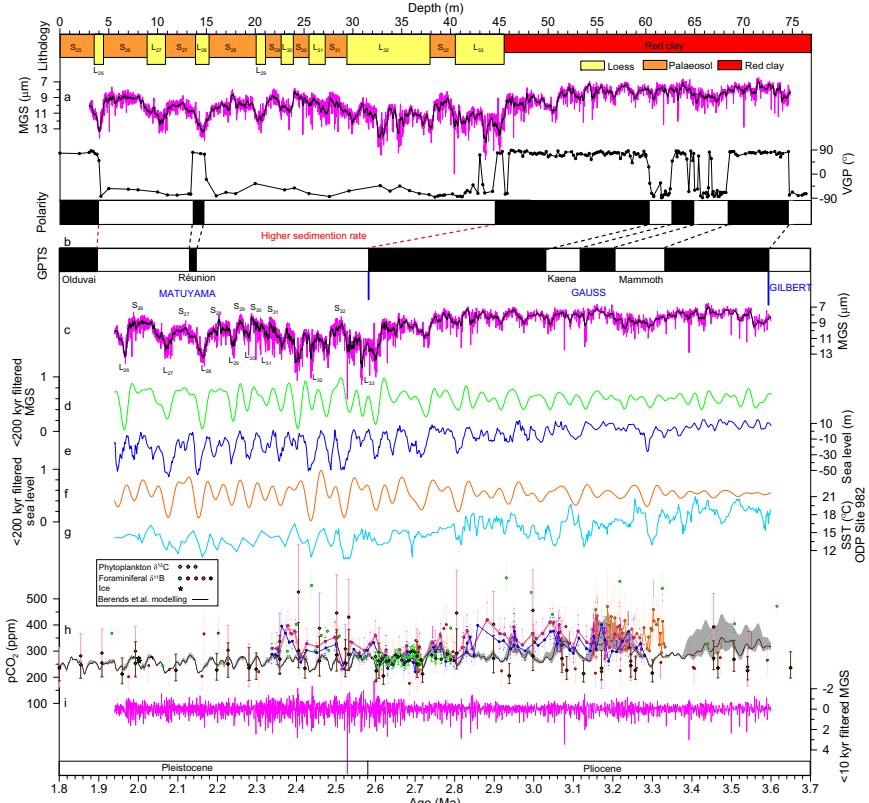

**Fig. 2 | Late Pliocene–early Pleistocene Asian winter monsoon and global climate changes. a** Lithology, original (magenta curve) and seven-point running average (black curve) mean grain size (MGS), virtual geomagnetic pole (VGP) latitude, and polarity zones for the Chongxin section plotted versus depth. S-numbers and L-numbers refer to consecutive palaeosol and loess horizons, respectively, counting back from the present-day. Black and white intervals represent normal and reverse polarity magnetozones, respectively. **b** Geomagnetic polarity time scale (GPTS)[39]. **c** Original (magenta curve) and seven-point running average (black curve) MGS time series for the Chongxin section. **d** Orbital (<200 kyr) Asian winter monsoon variability filtered from the original MGS series for the Chongxin section. **e** Global mean sea level reconstruction[5] and (**f**) its filtered orbital (<200 kyr) variability. **g** Sea surface temperature (SST) at ODP Site 982, North Atlantic Ocean[55,56]. **h** Atmospheric $CO_2$ reconstructions from boron isotopes (magenta[3], orange[8], green[9], blue[10], and red[47] circles), alkenones (pink[48], green[49], and brown[11] diamonds), Antarctic ice (green stars)[50], and inverse forward modelling (black curve)[7] (all with uncertainty bars/envelopes, see above references for detailed uncertainty definitions and calculations in various $CO_2$ reconstructions). **i** Millennial-scale AWM variability indicated by the <10 kyr filtered Chongxin MGS time series.

rates (Fig. 2a, b), and ASM weakening as indicated by distinct decreases in magnetic susceptibility[23], rates of chemical weathering[19,46], and soil carbonate $\delta^{13}C$ values[19]. AWM strengthening and ASM weakening were likely results of the colder and more intensely glaciated Pleistocene[5] associated with lower atmospheric $CO_2$ concentrations[3,7–11,47–50] (Fig. 2e–h).

In addition to orbital variability, previous high-resolution CLP loess-palaeosol grain size records spanning the past 1.5 Myr also suggest superposed millennial and even centennial variations[13,33,51,52]. Millennial and centennial oscillations in these younger records have lower amplitude than the orbital variability but higher amplitudes than analytical uncertainties associated with grain size measurements[13,33,51,52] (Supplementary Fig. 10). Our centennial-resolution MGS record reveals that lower-amplitude millennial oscillations were also superposed on higher-amplitude orbital-scale AWM variability between 3.6 Ma and 1.9 Ma, i.e., during the late Pliocene–early Pleistocene and across the iNHG (Fig. 2c), similar to what was previously recognized for the time interval spanning the last 1.5 Myr[13,33]. Expression of millennial-scale climate variability becomes more distinct in the <10 kyr filtered MGS record (Fig. 2i and Supplementary Fig. 10; "Methods"). As is the case for increased orbital amplitudes, the <10 kyr filtered MGS record is also marked by increased millennial-scale cycle amplitudes after ~2.66 Ma (Fig. 2i). Wavelet and power spectral analyses of the <10 kyr filtered MGS record reveal statistically significant millennial peaks against a red noise

model primarily between the 7-kyr and 1-kyr frequencies (Fig. 4). We note that millennial-scale age model uncertainties, which are linked to small uncertainties in geomagnetic boundaries and orbital tuning, sampling resolution, and smoothing related to pedogenesis, may have influenced expression of exact millennial periodicities. However, the combined effect of these factors was not strong enough to suppress the prevailing millennial variability in our centennial-resolution MGS record.

## Orbital-scale AWM dynamics

Our AWM proxy record, combined with a published global mean sea level record, indicates that ~100-kyr variability was present across the iNHG (Fig. 3), comparable to the interval after the MPT as recognized previously[34,35,37]. Consistent variation in orbital-scale global mean sea level and Chongxin MGS periodicities and amplitudes across the iNHG suggests that the NH ice volume forcing of orbital-scale AWM variability observed during the last 1.6 Myr[30,31,37] can be extended back to the late Pliocene–early Pleistocene. Generally, atmospheric $CO_2$ concentrations varied in pace with global mean sea level and temperature records on orbital timescales, which is well documented by proxy reconstructions and inverse forward modeling outputs for the last 800 kyr[5,7,53,54]. $CO_2$ is the principal greenhouse gas that amplified Earth's climate response to orbital forcing and largely determined Earth's thermal state. In addition to previously recognized NH ice volume forcing, orbital-scale atmospheric $CO_2$ forcing is also a key factor for

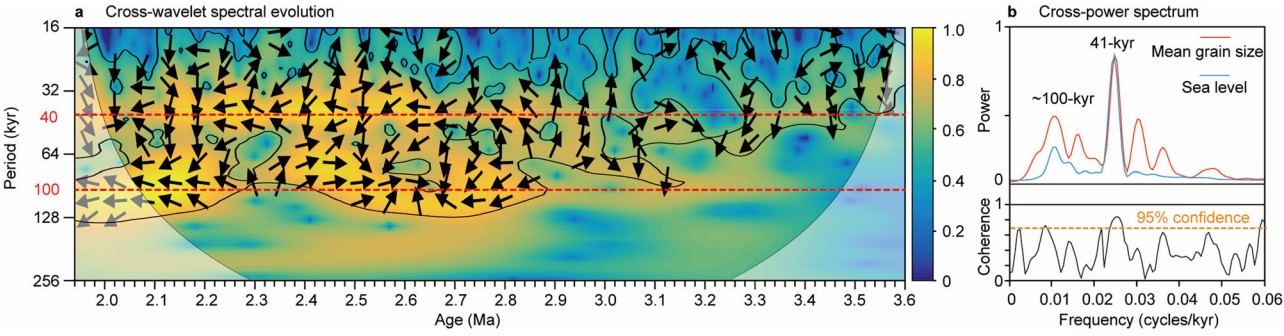

**Fig. 3 | Late Pliocene–early Pleistocene orbital climate variability. a** Cross-wavelet spectral evolution and (**b**) cross-power spectrum between the global mean sea level and the seven-point running average of the Chongxin mean grain size (MGS) record. Black contours in the cross-wavelet spectral evolution indicate the 5% significance level.

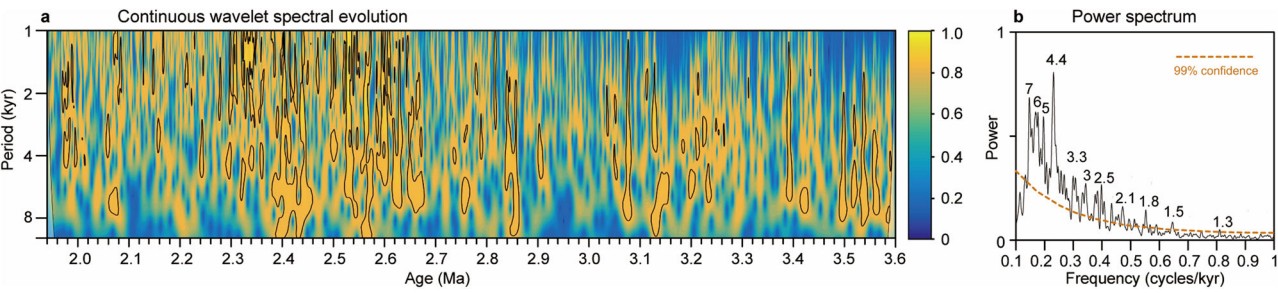

**Fig. 4 | Late Pliocene–early Pleistocene millennial AWM variability. a** Continuous wavelet spectral evolution and (**b**) power spectrum for the detrended Chongxin mean grain size (MGS) record (after removal of >10 kyr signals). Black contours in the wavelet spectral evolution indicate the 5% significance level.

Plio-Pleistocene AWM evolution. Continuous, orbitally resolved proxy-based $CO_2$ reconstructions from 3.6 to 1.9 Ma are currently unavailable. However, inverse forward modeling $CO_2$ estimates spanning the past 3.6 Myr[7] and shorter proxy-based $CO_2$ reconstructions spanning from 3.6 to 1.9 Ma[3,8–11,47–50] indicate that atmospheric $CO_2$ concentrations have similar orbital periodicities across the iNHG as global mean sea level and North Atlantic Ocean sea surface temperature records (Fig. 2e–h). This suggests that although various $CO_2$ reconstructions/model outputs may differ slightly from each other, it is plausible that strong coupling between climate and $CO_2$, as observed for the middle and late Pleistocene, was also a feature of late Pliocene–early Pleistocene climate. Glacial $CO_2$ lowering across the iNHG is distinct in both proxy and inverse forward modeling reconstructions (Fig. 2h). We infer that combined ice volume and radiative forcing (through both insolation and atmospheric $CO_2$ concentration changes) governed late Pliocene–early Pleistocene orbital-scale AWM variability, likely via both regional and global temperature modulations. The striking match between (1) a high-resolution sea surface temperature record from the North Atlantic Ocean[55,56] (Fig. 2g), (2) a global mean sea level record[5] (Fig. 2e), and (3) our grain size records over glacial-interglacial cycles across the iNHG (Fig. 2c), is consistent with the global ice volume/$CO_2$ forcing scenario. Moreover, typical glacial values shifted substantially in all three records across the iNHG, with lower sea surface temperatures and global mean sea levels corresponding to larger grain sizes (Fig. 2). Generally, larger ice sheets across the iNHG decreased Asian winter temperatures through increased albedo and atmospheric and oceanic circulation changes[30,31,44,57], while contemporaneous lower atmospheric $CO_2$ concentrations decreased Asian winter temperatures via weakened (regional) radiative forcing. This led to lower Asian winter temperatures, and, hence, enhanced the Siberian High pressure cell. In turn, this would have led to stronger AWM circulation during the colder Pleistocene. A marked stepwise glacial AWM intensity increase across the iNHG suggests a threshold AWM response to the more gradually increasing NH land-ice volume and decreasing global

atmospheric $CO_2$ concentration (Fig. 2). After the ice volume/$CO_2$ threshold was passed across the iNHG, the glacial AWM thus switched modes to a more intense circulatory pattern.

Glacial AWM intensification across the iNHG is also suggested by several CLP grain size records from the east and south of our study site[23,24] (Supplementary Fig. 11c–f). However, orbital AWM variability is not expressed as clearly in these records as in our centennial-resolution Chongxin grain size record, possibly because of their lower temporal (sampling) resolutions. Especially the persistence of orbital AWM variability from 3.6 to ~2.7 Ma is distinct in our record but is largely subdued or absent in previous records. Hence, we demonstrate that the AWM had a distinct orbital variability from 3.6 Ma onward, in dynamic response to a combination of ice volume and atmospheric $CO_2$ forcing. Consistent with global climatic trends and evolution, our higher resolution MGS record also indicates that glacial-interglacial AWM amplitudes were relatively lower before the iNHG.

**Millennial AWM dynamics**

Our high-resolution Chongxin grain size record is marked by persistent millennial variability superposed on orbitally-paced AWM variability throughout the late Pliocene–early Pleistocene (Figs. 2 and 4). It constitutes the oldest evidence reported for millennial AWM variability, more than two million years older than recent observations from other CLP records (<1.5 Ma)[13,58]. This finding is important for understanding millennial-scale climate dynamics in general (i.e., the Asian monsoon system is an integral part of the global climate system), because we document prevailing millennial AWM variability in both the colder, lower-$CO_2$ Pleistocene state and the more moderate glacials and warmer interglacials of the higher-$CO_2$ Pliocene world. We infer that millennial AWM variability can exist across a much broader range of climate-cryosphere boundary conditions than previously recognized, and may be a pervasive, long-term feature intrinsically linked to orbital-scale variability in the geological past[13–16,59–61]. For example, Tarim Basin loess records suggest that lower-amplitude

millennial-scale oscillations are superposed on larger-amplitude orbital variability in Westerly winds over the last 3.6 Myr[62]. Similar intriguing co-occurring millennial- and orbital-scale variability exists in multiple North Atlantic Ocean records during the last 3.2 Myr[14,59].

Modern observations and model simulations suggest that abrupt AWM intensity changes are closely related to high-latitude NH temperature changes[63–65] (Fig. 1a). Similarly, millennial AWM variations were probably linked to NH temperature changes between 3.6 and 1.9 Ma, and influenced by (1) ice sheet dynamics, (2) oceanic meridional overturning circulation, (3) radiative forcing, and/or (4) combination tones and harmonics of primary orbital climate cycles[14,16,60,61,66–74]. Model simulations suggest that independent minor changes in NH ice sheet thickness, greenhouse gas concentrations, and insolation can trigger prominent millennial-scale NH climate variations through sea ice-ocean-atmosphere interactions or Atlantic meridional overturning circulation (AMOC)[66–70]. Carbon and neodymium isotopic compositions of North Atlantic Ocean sediments suggest distinct late Pliocene–early Pleistocene millennial AMOC oscillations[75]. Combination tones and harmonics of Milankovitch cycles can also induce millennial-scale NH climate variability[60,61,72–74], especially on multi-millennial time scales. Similar mechanisms probably governed late Pliocene–early Pleistocene millennial AWM variability. We infer that both internal climate dynamics (ice sheet dynamics, ocean-atmosphere condition changes, and combination tones and harmonics of primary astronomical cyclicity) and external astronomical forcing (insolation) may have contributed to the observed late Pliocene–early Pleistocene millennial AWM variability by modulating high-latitude Asian winter temperature.

Our unique centennial-resolution 3.6–1.9 Ma grain size record from a representative aeolian red clay and loess-palaeosol sequence from the central CLP provides critical insights into both orbital- and millennial-scale monsoon variability during a key interval from the late Pliocene to early Pleistocene and across the iNHG. NH ice sheet expansion and atmospheric $CO_2$ decrease enhanced glacial AWM intensity but did not shift its orbital-scale periodicities across the iNHG. Both 41-kyr and ~100-kyr cycles existed in the late Pliocene–early Pleistocene AWM in response to both ice volume and atmospheric $CO_2$ drivers. Superposed on orbital-scale changes in our centennial-resolution grain size record are millennial AWM variations. We show that millennial AWM variability persisted during both glacials and interglacials of the warmer late Pliocene–early Pleistocene. Millennial AWM oscillations thus occurred across a broader range of climate-cryosphere boundary conditions and more than two million years earlier than previously recognized. These millennial AWM variations probably constitute a long-term, pervasive feature of monsoon-dominated (hydro-) climate variability that was governed by both external astronomical forcing and internal Earth climate dynamics.

## Methods
### Sampling
After cleaning and removal of surface outcrop, we collected 3571 unoriented samples from the Chongxin section, central CLP, at 2 cm intervals (equivalent to an average time spacing of ~0.5 kyr) for grain size analyses. For magnetostratigraphic analysis, we also collected 251 parallel block samples that were oriented in the field with a compass. Subsequently, oriented block samples were cut into 2 cm × 2 cm × 2 cm cubic samples in the laboratory for thermal demagnetization to establish a magnetochronology. Remaining material from oriented block samples was used for mineral magnetic measurements. All magnetic and grain size analyses were conducted at the Institute of Earth Environment, Chinese Academy of Sciences, Xi'an.

### Grain size analysis
Grain size distributions of red clay and loess-palaeosol samples were measured in the laboratory. Before the laser diffraction

measurements, 0.2–0.3 g of each sample was pretreated with 30% $H_2O_2$ and 10% HCl to remove organic matter and carbonates, respectively. Treated samples were then put into 10 ml 10% $(NaPO_3)_6$ solution in an ultrasonic bath for ~10 min for dispersion. After sample pretreatment, grain size was analyzed in 101 size bins using a Malvern 3000 Laser Instrument with a Hydro LV wet dispersion unit from 0.02 μm to 2000 μm. Light sources include a red He–Ne laser at a 623 nm wavelength and a blue light emitting diode at 470 nm. Constants of 1.33 for the refractive index of water, 1.52 for the refractive index of solid phases, and 0.1 for the absorption index were used. We maintained a pump speed of ~2900 rpm in the Hydro LV pump. Each sample was measured three times. Grain size data were processed with the Malvern Mastersizer 3000 software (version 3.81), which transforms scattered light data to particle size information based on the Mie Scattering Theory.

### Magnetic analyses
To identify the magnetic minerals that record the palaeomagnetic signal, we measured high-temperature-dependent magnetic susceptibility (χ–T), isothermal remanent magnetization (IRM) acquisition, magnetic hysteresis loops, and first-order reversal curve (FORC) diagrams for six red clay/loess-palaeosol samples from the Chongxin section. χ–T curves were measured in an argon atmosphere from room temperature to 700 °C and back to room temperature using a MFK1-FA magnetic susceptibility meter equipped with a CS-3 high-temperature furnace (AGICO, Brno, Czech Republic). IRM acquisition curves, magnetic hysteresis loops, and FORC diagrams were measured with a Princeton Measurements Corporation (Model 3900) vibrating sample magnetometer (VSM). Each IRM acquisition curve contains 200 data points that were measured at logarithmically spaced field steps to 1 T. Hysteresis loops were measured to ±1 T or ±1.5 T at 3 mT increments, with a 300 ms averaging time. First-order reversal curves (FORCs) were measured at 5 mT increments to ~600 mT, with a 100 ms averaging time. FORC data were processed using the FORCinel package[76], with a smoothing factor of 3.

All measured χ–T heating curves are characterized by a major χ decrease near 580 °C (Supplementary Fig. 2), which indicates that magnetite is the magnetically dominant phase in the sediment. Most samples indicate a steady χ increase below 300 °C during heating, which may be caused by gradual unblocking of very fine-grained ferrimagnetic (i.e., the superparamagnetic (SP) and fine-grained single-domain (SD)) particles, or stress release upon heating in such particles[77–79]. All samples have a subsequent χ decrease between ~300 and ~500 °C, which is consistent with conversion of ferrimagnetic maghemite to weakly magnetic hematite[80–82]. In agreement with a dominant low-coercivity magnetite/maghemite contribution, all IRM acquisition curves have a major increase below 300 mT (Supplementary Fig. 3), and most hysteresis loops close below ~300 mT (Supplementary Fig. 4). All FORC diagrams have closed contours with maximum contour density at coercive force ($B_c$) values <20 mT (Supplementary Fig. 5), which suggests a substantial presence of SD magnetite[83,84]. Outer contours are generally divergent along the $B_u$ axis, which suggests the presence of small vortex state magnetic particles[85]. A general vertical FORC distribution immediately adjacent to the $B_u$ axis in the lower half plane is indicative of SP particles[83,85–88]. The hematite expression in the rock magnetic results is not distinct because its presence is generally masked magnetically by the much stronger magnetization of relatively smaller amounts of magnetite[25]. However, substantial amounts of pigmentary red hematite particles are suggested by red hues in the loess-palaeosol and red clay sediments (Supplementary Fig. 1).

### Palaeomagnetic analyses
To establish a clear palaeomagnetic polarity sequence for the Chongxin section, 251 oriented samples were subjected to detailed

stepwise thermal demagnetization of the natural remanent magnetization (NRM), which was conducted using a TD-48 thermal demagnetizer. Samples were heated at 14–15 successive steps with 20–50 °C increments to a maximum temperature of 600–620 °C, at which point >90% of the initial NRM was demagnetized. After each demagnetization step, the remaining NRM was measured with a 2-G Enterprises Model 755-R cryogenic magnetometer housed in a magnetically shielded space. Demagnetization results were evaluated using orthogonal diagrams[89]; the principal component direction for each sample was computed using least-squares linear fitting[90]. Principal component analysis (PCA) was performed with PaleoMag software[91]. More samples (196) were demagnetized for the red clay sequence than the loess-palaeosol sequence (55 samples) because (1) the former (red clay) has lower sedimentation rates and contains several short-duration magnetic polarity zones, and (2) geomagnetic polarity reversals have been identified more precisely in the latter (loess-palaeosol) in previous studies[20,21,23,24,92].

The NRM contains a secondary overprint that was removed by thermal demagnetization to 200–300 °C, followed by isolation of the characteristic remanent magnetization (ChRM) up to 600–620 °C (Supplementary Fig. 6). At least four consecutive demagnetization steps that decay linearly to the origin of orthogonal diagrams were used to determine ChRM directions above 250–350 °C, with a maximum angular deviation (MAD) ≤15° for line fits (not anchored to the origin)[93]. From 251 demagnetized samples, 212 yielded stable ChRM directions, from which virtual geomagnetic pole (VGP) latitudes were calculated to establish the magnetostratigraphic zonation. The Chongxin section records five normal and five reverse polarity zones (Fig. 2a and Supplementary Fig. 7b). Each zone is defined based on at least three consecutive VGP latitudes of identical polarity. Combining the well-established age of the iNHG for the boundary between loess-palaeosol and red clay on the CLP, we can readily correlate the Chongxin magnetostratigraphy to the geomagnetic polarity timescale (GPTS)[39]. The section spans from the youngest Gilbert reverse polarity chron to the Olduvai normal subchron, with a straightforward magnetostratigraphic correlation to the late Pliocene–early Pleistocene GPTS (Fig. 2a, b and Supplementary Fig. 7a, b). Our magnetostratigraphic assignment for the Chongxin section is consistent with those of other central and eastern CLP red clay/loess-palaeosol sections, such as at Jingchuan[20,24], Xifeng[23], Lingtai[21], Baoji[20,24], Lantian[24], Shilou[25], and Jiaxian[26] sections (Supplementary Fig. 7).

### Age model development

We combine magnetostratigraphy and astronomical tuning to establish an age model for the Chongxin section. Magnetostratigraphy was used to establish a first-order age model between 3.6 and 1.9 Ma for the Chongxin section based on linear interpolation between subsequent tie points using the nine identified geomagnetic polarity reversals, which were assigned ages from the GPTS[39] (Supplementary Fig. 8a, b and Supplementary Table 1). Using this magnetochronology, the Chongxin MGS record has distinct ~100-kyr eccentricity and 41-kyr obliquity bands between 3.6 and 1.9 Ma (Supplementary Fig. 9a). Accordingly, we refined the magnetochronology by tuning the high-frequency 41-kyr MGS variations to orbital obliquity cycles in the astronomical solution[94] with an automatic orbital tuning procedure[43]. Following previous orbital tunings of the CLP loess-palaeosol sequences with this procedure[20,23], we repeatedly matched the 41-kyr cycles filtered from the Chongxin MGS record with the 8-kyr-lagged obliquity curve by manually adding time control points. An 8 kyr lag between the AWM and obliquity is used commonly in tunings of CLP loess-palaeosol grain size records[20,23]. Generally, larger MGS and filtered 41-kyr grain size peaks are associated with strong AWM and are correlated with obliquity minima. Ages for palaeomagnetic reversals are not kept fixed to optimize tuning results given uncertainties in palaeomagnetic boundaries and post-depositional NRM lock-in depth

in aeolian sediments[95,96], whereas tuned ages should not differ much (less than two obliquity cycles: <80 kyr) from those obtained from the palaeomagnetic record. After iteratively adding and/or adjusting age control points, the most likely astronomical timescale is shown in Supplementary Fig. 8c. The astronomical timescale is constrained by 31 tie points where MGS maxima facilitate tie point selection throughout the interval (Supplementary Fig. 8c and Supplementary Table 2). The filtered 41-kyr grain size component essentially correlates cycle-to-cycle with the calculated orbital obliquity in both coherency and amplitude modulation patterns in the astronomical timescale (Supplementary Fig. 8c). Palaeomagnetic reversal ages in the astronomical timescale are broadly consistent with their GPTS ages (Supplementary Table 1). Good phase matching is a function of tuning, while amplitude matches are independent of tuning, which supports the accuracy of our astronomical age model[20,23]. Overall, differences between the magnetochronology and our astronomical timescale obtained from obliquity tuning are not significant. The MGS record has a broadly similar orbital expression in our astronomical and magnetostratigraphic age models, with distinct co-occurring eccentricity and obliquity cycles (Supplementary Fig. 9). Orbital tuning refines the age model, corrects displaced obliquity cycles in the untuned magnetostratigraphic age model, and enhances the recorded orbital expression, but it does not change primary orbital periodicities. Therefore, the MGS record has higher obliquity power and lower non-orbital noise in our refined astronomical timescale than in the untuned magnetostratigraphic age model (Supplementary Fig. 9).

### Spectral analyses

We calculated cross-wavelet and cross-power spectra between the global mean sea level and the seven-point running average of the Chongxin MGS record using the *Acycle* software[97] to better evaluate late Pliocene to early Pleistocene orbital variability. To improve expression of orbital-scale variability in global mean sea level and the seven-point running average of the Chongxin MGS record, their <200 kyr variability was filtered. To improve expression of millennial variability in the Chongxin MGS record, <10 kyr variability was filtered. A high-pass filter in the *Acycle* software[97] was used to obtain the <200 kyr and <10 kyr components. This high-pass filtering isolates higher frequency signals and excludes the influence of lower frequency signals. The <10 kyr filtered MGS record was also used for wavelet power spectral evolution and power spectrum analyses. Wavelet power spectral evolution was calculated using the *Acycle* software[97]. We used the 2π-Multi-Taper Method (MTM) to analyze power spectra with the function Spectral Analysis[98].

### Reporting summary

Further information on research design is available in the Nature Portfolio Reporting Summary linked to this article.

### Data availability

All data that support the findings of this research are provided in the East Asian Paleoenvironmental Science Database http://paleodata.ieecas.cn/FrmDataInfo_EN.aspx?id=81837a9f-33af-4feb-916f-c9e09c4a9f53.

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

## Acknowledgements

We thank professor Zhisheng An and Dr. A.C. Da Silva for helpful discussions. This study was supported financially by the Chinese Academy of Sciences Strategic Priority Research Program (XDB 40000000 to H.A.), the Second Tibetan Plateau Scientific Expedition and Research Program (2019QZKK0707 to H.A.), the Chinese Academy of Sciences Key Research Program of Frontier Sciences (QYZDB-SSW-DQC021 to H.A.), the National Natural Science Foundation of China (42074076 to H.A.), the Fund of Shandong Province (LSKJ202203300 to H.A. and P.Z.), the Shaanxi Province Youth Talent Support Program (to H.A.), and the Australian Research Council (DP200100765 to A.P.R.).

## Author contributions

H.A. conceived the study. H.A., Y.S., X.Q., and P.Z. contributed to sampling and magnetic and grain size measurements. X.L. conducted spectral analysis. D.L., M.J.D., A.P.R., Z.J., T.J., Q.L., Q.S., and C.H. contributed to proxy analysis, interpretation, and discussion. H.A. led manuscript writing with intellectual contributions from D.L., M.J.D., A.P.R., and T.J.

## Competing interests

The authors declare no competing interests.
