## [Peer Review File · Nature Communications]

Orbital- and millennial-scale Asian winter monsoon variability across the Pliocene–Pleistocene glacial intensificationREVIEWER COMMENTS

Reviewer #1 (Remarks to the Author):

Review of the manuscript "Orbital and millennial Asian monsoon variability across the Pliocene–Pleistocene glacial intensification" by Ao et al. [NCOMMS-23-40160]

General comments:

The intensification of Northern Hemisphere glaciation across the Pliocene–Pleistocene transition marks the final step in the overall cooling trend to a fully established bipolar icehouse. As a key component of the global climate system, the Asian monsoon has been extensively investigated for both the Pleistocene and the Pliocene; however, its response to such a milestone event in the Earth's Cenozoic climate history on different timescales remains unclear. In this study, the authors document a complete evolution of the Asian winter monsoon (AWM) variability on both orbital and millennial scales for the period from the late Pliocene to the early Pleistocene based on a continuous, centennial-resolved grain-size record from a new loess-paleosol/red clay section on the central Chinese Loess Plateau. They suggest that intensification of Northern Hemisphere glaciation had not altered orbital-scale AWM periodicities but could have enhanced both glacial and interglacial AWM intensities; while millennial AWM intensity fluctuations persisted during both the warmer late Pliocene and the colder early Pleistocene. The data and inferences presented in this study are of great significance and would contribute to a deep understanding of the mechanism responsible for Asian monsoon variability on millennial scale. I recommend acceptance of this manuscript for publication in Nature Communications after minor revisions.

Specific comments:

1. Lines 68–70: There is no logic in this argument. That is to say, lacking a hydrological signature that marks ASM records cannot be the reason for more difficulties to reconstruct AWM variability. This is because AWM is distinct in nature from ASM as explained on lines 56–63.
2. Lines 139–142: Note that the differences should be manifested in both the magnitude of glacial–interglacial variations and the minima of low interglacial values.
3. Lines 153–155: How young is younger and how old is older? Are the young / the old younger / older than the period of the iNHG? It is better to say specially.
4. Lines 201–208: It appears better to move these sentences to 'Introduction'.
5. Lines 240–293: It appears better to move this part to before 'Millennial-scale AWM dynamics'.

Minor comments:

1. Line 30: Change 'palaeosol' (in the entire manuscript) to 'paleosol'.
2. Line 72: Change 'central' to 'north-central'.
3. Line 96: Change 'accumulated' to 'is covered with'.
4. Line 103: Change 'variability' to 'variations'.
5. Line 105: Delete 'from this section'.
6. Line 106: Delete 'first-order'.
7. Line 113: Change 'record' to 'sequence'.
8. Line 119: Insert 'and' before 'in the lower ...'.
9. Line 122: Change 'variable sampling data' to 'different sampling intervals'.
10. Line 124: Delete 'magnetostratigraphy'.
11. Line 128: Change 'However' to 'although' and merged this sentence into the one before it.
12. Line 131: Change 'records' to 'sequences'.
13. Line 135: Delete 'further'.
14. Lines 143–144: Delete 'that is underpinned by a detailed initial magnetostratigraphic age model'.
15. Lines 152–153: Change 'a greater aeolian dust transport capacity by' to 'drier conditions in the dust source regions as well as'.
16. Lines 156–159: This sentence needs to be rewritten. It is not clear.
17. Lines 161–162: Change 'an observation that provides' to 'providing'.

18. Line 164: Change 'thick higher' to 'thicker'.
19. Line 164: Insert 'relatively thicker' to before 'yellow'.
20. Lines 162–168: Both sentences sound repeated.
21. Line 180: Delete 'Visual inspection of'.
22. Line 185: Delete 'Furthermore'.
23. Line 197: insert 'strong' between 'not' and 'enough'.
24. Lines 209–210: Change the passive voice to the active one.
25. Line 212: insert 'a' between 'such' and 'AWM'.
26. Line 213: Change 'now' to 'in this study'.
27. Line 217: Delete 'central Asian'.
28. Line 243: Delete 'both'.
29. Line 246: Insert 'and' between 'volume' and 'atmospheric'.
30. Line 264: Delete 'sheet'.
31. Line 266: Change 'Both ice volume and greenhouse gas concentrations' to 'These two factors'.
32. Line 286: Change 'sheet' to 'volume' (two places).
33. Line 292: Change 'a pervasive and long-term' to 'a long-term, pervasive'.

Reviewer #2 (Remarks to the Author):

Please see attached pdf file

Reviewer #3 (Remarks to the Author):

Review of „Orbital and millennial Asian monsoon variability across the Pliocene–Pleistocene glacial intensification” by Ao et al.

The East Asian winter and summer monsoon (EAWM/EASM) affects the lives of millions of people, so understanding how the EAWM/EASM operated in the past is important in a warming climate that now more and more resembles conditions at the end of the Pliocene in terms of both CO₂ levels and average temperatures. Ao and co-workers present a new, high resolution grain size record (representing EAWM) with a strong paleomagnetic age control from the central part of the Chinese Loess Plateau, spanning a period of 3.6-1.9 Ma (late Pliocene/early Pleistocene). They find that late Pliocene–early Pleistocene EAWM is characterized by persistent 41-kyr and ~100-kyr cycles, and superimposed on orbital variability, millennial timescale EAWM fluctuations occurred much earlier than previously thought. The data presented in the manuscript reflect the results of very significant field and laboratory work, with the authors performing paleomagnetic measurements of 251 samples and laser diffraction analysis of 3487 loess/paleosol/red clay samples, which makes this study impressive and unique. The text of the manuscript is very well written, clear and easy to follow, and most of the authors' conclusions are logical and scientifically sound. There are of course some issues that need to be clarified in a minor revision, but in my opinion (with perhaps one exception) they are not of such importance that they would fundamentally affect the main conclusions of the paper. My concerns are primarily related to chapter "Orbital-scale AWM dynamics" (lines 273-279) and Figure 2, and I think the text needs some minor amendments:

- 1) From Figure 2c, it can be seen that interglacial intensities were similar before / after the iNHG or perhaps a little less intense after 2.7 Ma. As for the glacials, they clearly became more intense after 2.7 Ma. So, the statement that both glacials and interglacials became more intense ("marked stepwise increase in both glacial and interglacial AWM intensity") is not evident in the MGS record, only that glacials became more intense and the amplitudes of fluctuations between glacials/interglacials. This must be addressed during the revisions.
- 2) Based on Fig. 2, the MGS record resembles the sea level curve, with larger amplitude fluctuations after 2.7 Ma. This is in marked contrast to the CO₂ record showing larger (sometimes >100 ppm)

swings before the iNHG, while smaller scale fluctuations (~40 ppm) after 2.7 Ma. At the same time, CO₂ concentrations fluctuated around a lower mean value (~260 ppm) after the iNHG indicating significant changes in the GHG boundary conditions (likewise ice volume), leading to a different climate state. Is this what the authors are referring to?

Other, less important questions also arise in relation to the interpretations or methodology:

3) This is just a pedantic point, but the authors describe in several places (e.g. lines 68, 83) that the record they examine is continuous. Unfortunately, this is only an assumption. It would be difficult to imagine a dynamic eolian environment such as the CLP without shorter/longer (up to at least decades or a few hundred years) erosional periods over the last 3.6 million years. There is direct evidence for sedimentation gaps at the CLP margin based on luminescence dating results (see Stevens et al., 2018 and references therein). Therefore, I suggest using the term "quasy-continuous" in the text.

4) The authors state that "Mean grain sizes of bulk samples and extracted quartz particles have almost identical variation for both red clay and loess-palaeosol sequences on the CLP", by referring to Sun et al. (2010). In my view, this should have been tested using a subset of their samples to prove that this is true at the study site. For instance, bulk and quartz grain size variations were shown to be different in European loess (Újvári et al., 2016, 2017). I'm not stating that this holds true for the CLP, but it would have been worth testing.

5) Line 151-153, "coarser grain size-stronger winds relation": I agree that stronger wind is one of the most likely explanations, in addition to other factors that influence the transport capacity of the medium, including, for example, the state of the eolian surface and temperature of the transport medium. At lower temperatures an aeolian surface is more active, cohesive forces are weaker and the air is denser, so under colder conditions larger particles can be transported (see 3.1.1.4 subchapter in Ujvari et al., 2016).

6) Methods section, grain size analyses (lines 305-309): Provide information on instrument parameters (absorption and refractive index). Have you used the Mie theory (or Fraunhofer approximation) for GSD calculations?

7) Methods section, age model (lines 377-379): Orbital tuning may have resulted in increased power at the 41 ka band, nicely seen in Fig S9. Could you comment on this?

8) Methods section, 8 ka lag (lines 381-382): This lag is debated, others argue for a shorter lag (~5 ka) based on absolute ages (Stevens et al., 2018). However, the ~ 5 ka lag was proposed for the response of EASM to insolation forcing.

9) Methods, age model (lines 386): What was the accepted tolerance of paleomagnetic boundary shifts in kyrs?

Technical corrections, typos

line 103: "unprecedented" should be replaced by "unprecedented"

line 137: write "AMW indicator"

line 192: consider writing "2.66 Ma" instead of "2.7 Ma"

line 245: "CO₂ concentration increase", do you mean "decrease"?

line 246: write "ice volume and atmospheric CO₂"

lines 259-260: consider using "different control" instead of "larger control"

line 374: write "which were assigned"

line 641: replace "Loss" by "Loess"

Figure 1: I suggest changing the label of Fig. 1b to make sure these are readable. What does the hatching mean?

References

Stevens, T., Buylaert, JP., Thiel, C. et al. (2018). Ice-volume-forced erosion of the Chinese Loess Plateau global Quaternary stratotype site. *Nature Communications* 9, 983.

Sun, Y.B., An, Z.S., Clemens, S.C., Bloemendal, J., Vandenberghe, J. (2010). Seven million years of wind and precipitation variability on the Chinese Loess Plateau. *Earth and Planetary Science Letters* 297, 525–535.

Újvári, G., Kok, J.F., Varga, G., Kovács, J. (2016). The physics of wind-blown loess: implications for grain size proxy interpretations in Quaternary paleoclimate studies. *Earth-Science Reviews* 154, 247–278.

Újvári, G., Stevens, T., Molnár, M., Demény, A., Lambert, F., Varga, G., et al. (2017). Coupled European and Greenland last glacial dust activity driven by North Atlantic climate. *Proceedings of the National Academy of Sciences of the United States of America*, 114, 10622–10631.

We thank the reviewers and the editor for evaluating our manuscript. Please find below our point-by-point responses in blue. Our revisions in the main text are also marked blue.

Reviewer #1

General comments:

The intensification of Northern Hemisphere glaciation across the Pliocene–Pleistocene transition marks the final step in the overall cooling trend to a fully established bipolar icehouse. As a key component of the global climate system, the Asian monsoon has been extensively investigated for both the Pleistocene and the Pliocene; however, its response to such a milestone event in the Earth’s Cenozoic climate history on different timescales remains unclear. In this study, the authors document a complete evolution of the Asian winter monsoon (AWM) variability on both orbital and millennial scales for the period from the late Pliocene to the early Pleistocene based on a continuous, centennial-resolved grain-size record from a new loess-paleosol/red clay section on the central Chinese Loess Plateau. They suggest that intensification of Northern Hemisphere glaciation had not altered orbital-scale AWM periodicities but could have enhanced both glacial and interglacial AWM intensities; while millennial AWM intensity fluctuations persisted during both the warmer late Pliocene and the colder early Pleistocene. The data and inferences presented in this study are of great significance and would contribute to a deep understanding of the mechanism responsible for Asian monsoon variability on millennial scale. I recommend acceptance of this manuscript for publication in Nature Communications after minor revisions.

Response: We thank Reviewer #1 for positive evaluation and helpful comments.

Specific comment #1

Lines 68–70: There is no logic in this argument. That is to say, lacking a hydrological signature that marks ASM records cannot be the reason for more difficulties to reconstruct AWM variability. This is because AWM is distinct in nature from ASM as explained on lines 56–63.

Response: Thank you. Indeed, in retrospect the argument did not make that much sense. We have removed this argument.

Specific comment #2

Lines 139–142: Note that the differences should be manifested in both the magnitude of glacial–interglacial variations and the minima of low interglacial values.

Response: We have included this note in the revised manuscript (lines 156-157).

Specific comment #3

Lines 153–155: How young is younger and how old is older? Are the young / the old younger / older than the period of the iNHG? It is better to say specially.

Response: These terms are now rephased more specifically. To avoid repetition, the former sentence on lines 153–155 has been deleted.

Specific comment #4

Lines 201–208: It appears better to move these sentences to ‘Introduction’.

Response: These sentences have been revised and moved to the “Introduction” as suggested (lines 62-68).

Specific comment #5

Lines 240–293: It appears better to move this part to before ‘Millennial-scale AWM dynamics’.

Response: We have implemented this change.

Minor comment #1

1. Line 30: Change ‘palaeosol’ (in the entire manuscript) to ‘paleosol’.

Response: “Palaeosol” and “paleosol” represent British and American English, respectively. Nature publications use British English, so we have not changed “palaeosol” to “paleosol” in the revised manuscript.

Minor comment #2–7

2. Line 72: Change ‘central’ to ‘north-central’.

3. Line 96: Change ‘accumulated’ to ‘is covered with’.

4. Line 103: Change ‘variability’ to ‘variations’.

5. Line 105: Delete ‘from this section’.

6. Line 106: Delete ‘first-order’

7. Line 113: Change ‘record’ to ‘sequence’.

Response: We have incorporated these suggested changes.

Minor comment #8

8. Line 119: Insert ‘and’ before ‘in the lower ...’.

Response: To clarify this sentence, we have added roman numerals to the list of positions where the Olduvai subchron is found (lines 128-131).

Minor comment #9–10

9. Line 122: Change ‘variable sampling data’ to ‘different sampling intervals’.

10. Line 124: Delete ‘magnetostratigraphy’.

Response: We have incorporated these suggested changes.

Minor comment #11

11. Line 128: Change ‘However’ to ‘although’ and merged this sentence into the one before it.

Response: We have rephrased related sentences in the revised manuscript (lines 137-140).

Minor comment #12–14

12. Line 131: Change ‘records’ to ‘sequences’.

13. Line 135: Delete ‘further’.

14. Lines 143–144: Delete ‘that is underpinned by a detailed initial magnetostratigraphic age model’.

Response: We have included these changes.

Minor comment #15

15. Lines 152–153: Change ‘a greater aeolian dust transport capacity by’ to ‘drier conditions in the dust source regions as well as’.

Response: We have changed it to “increased dust transport capacity by stronger AWM winds from drier source regions to the north and west of the CLP under colder conditions across the iNHG (Ding et al., 1995; Hao et al., 2012; Sun et al., 2010; Ujvári et al., 2016; Xiao et al., 1995)”.

Minor comment #16–17

16. Lines 156–159: This sentence needs to be rewritten. It is not clear.

17. Lines 161–162: Change ‘an observation that provides’ to ‘providing’.

Response: We have clarified these sentences.

Minor comment #18–20

18. Line 164: Change ‘thick higher’ to ‘thicker’.

19. Line 164: Insert ‘relatively thicker’ to before ‘yellow’.

20. Lines 162–168: Both sentences sound repeated.

Response: To avoid repetition, we have removed this sentence.

Minor comment #21–33

21. Line 180: Delete ‘Visual inspection of’.

22. Line 185: Delete ‘Furthermore’.

23. Line 197: insert ‘strong’ between ‘not’ and ‘enough’.

24. Lines 209–210: Change the passive voice to the active one.

25. Line 212: insert ‘a’ between ‘such’ and ‘AWM’.

26. Line 213: Change ‘now’ to ‘in this study’.

27. Line 217: Delete ‘central Asian’.

28. Line 243: Delete ‘both’.

29. Line 246: Insert ‘and’ between ‘volume’ and ‘atmospheric’.

30. Line 264: Delete ‘sheet’.

31. Line 266: Change ‘Both ice volume and greenhouse gas concentrations’ to ‘These two factors’.

32. Line 286: Change ‘sheet’ to ‘volume’ (two places).

33. Line 292: Change ‘a pervasive and long-term’ to ‘a long-term, pervasive’.

Response: We have adopted these helpful edits.

References cited in our response above

Ding, Z.L., Liu, T.S., Rutter, N.W., Yu, Z.W., Guo, Z.T. and Zhu, R.X., 1995. Ice-volume forcing of East Asian winter monsoon variations in the past 800,000 Years. *Quaternary Research*, 44(2): 149–159.

Hao, Q.Z., Wang, L., Oldfield, F., Peng, S.Z., Qin, L., Song, Y., Xu, B., Qiao, Y.S., Bloemendal, J. and Guo, Z.T., 2012. Delayed build-up of Arctic ice sheets during 400,000-year minima in insolation variability. *Nature*, 490(7420): 393–396.

Sun, Y.B., An, Z.S., Clemens, S.C., Bloemendal, J. and Vandenberghe, J., 2010. Seven million years of wind and precipitation variability on the Chinese Loess Plateau. *Earth and Planetary Science Letters*, 297(3-4): 525–535.

Ujvári, G., Kok, J.F., Varga, G. and Kovács, J., 2016. The physics of wind-blown loess: Implications for grain size proxy interpretations in Quaternary paleoclimate studies. *Earth-Science Reviews*, 154: 247–278.

Xiao, J.L., Porter, S.C., An, Z.S., Kumai, H. and Yoshikawa, S., 1995. Grain size of quartz as an indicator of winter monsoon strength on the loess plateau of central China during the last 130000 yr. *Quaternary Research*, 43: 22–29.

Reviewer #2

General comments:

This manuscript by Ao et al presents a new, continuous, extremely high-resolution dataset of grain size from a sedimentary section in the central Chinese Loess Plateau, interpreted to represent changes in dust delivery to the area by Asian winter monsoon winds. The record is of very high quality, allowing excellent paleomagnetic age control and orbital tuning. High-resolution, continuous records such as this one that span the intensification of northern hemisphere glaciation are precious in terms of our understanding of monsoon dynamics over periods of evolving global climate boundary conditions. The paper is very well written, well referenced, and clearly structured. The figures and supplementary figures are very nice (although I have a few suggestions for additions). I would like to state that I am not an expert on palaeomagnetism or loess records, so my review focuses on the more general monsoon aspects and the iNHG context. The new grain size record is clearly explained and interpreted; however, I think the paper would benefit from additional context and comparison with other summer/winter south Asian/east Asian monsoon records that span iNHG, and also a more thorough assessment of $p\text{CO}_2$ changes and their potential role on monsoon evolution. Below are some specific comments and thoughts that I hope will be constructive to the authors when revising the paper.

Response: We thank Reviewer #2 for their helpful feedback and comments on our manuscript. Following these suggestions, we include additional context and comparison with other summer/winter south Asian/east Asian monsoon records spanning the iNHG (lines 255-263; Supplementary Note 1), with a more thorough assessment of $p\text{CO}_2$ changes and their potential role on monsoon evolution (Fig. 2h and lines 213-243).

Specific comment #1

Line 34: in phase (rather than in pace)?

Response: It is not accurate to use “in phase” here because phases of orbital-scale oscillations of Asian winter monsoon, ice sheet, and atmospheric CO_2 concentration are not always identical, with slight lags and leads. We change “in pace with” to “in response to”, to clarify the sentence.

Specific comment #2

Line 43: ice-sheet size or volume.

Response: We checked this sentence: “Middle and Late Pleistocene” is more reasonable than “Middle and Late Pleistocene ice-sheet size or volume” (lines 45 in the revised manuscript).

Specific comment #3

Lines 44-48: It would make sense to specify/restrict these sentences to the mid-Pliocene warm period, if you want to talk about higher than pre-industrial $p\text{CO}_2$ and temperatures.

Response: Thanks, we have re-written these sentences (lines 49-54).

Specific comment #4

Line 52: long-term rather than overall?

Response: We adopted this change.

Specific comment #5

Line 80: I think it would be great to include somewhere (ideally in the main text) a plot of the new record with other AWM/ASM records that span iNHG. This will help highlight the novelty and added

value of the new record, as well as being useful to refer to in the discussion.

Response: We now compare other AWM/ASM records with our grain size record spanning the iNHG in Supplementary Fig. 11, with related discussion in the main text (lines 255-263) and Supplementary Note 1. Our study focuses on the AWM, so related discussion of ASM variability across the iNHG is included in Supplementary Note 1 to avoid distraction. However, the related other AWM records are discussed in the main text (lines 255-263).

Specific comment #6

Line 86: “we assess [...] through detailed land-sea comparison” – although this would be great, it isn’t really the case in the current version of the manuscript. Marine monsoon records over iNHG are not discussed.

Response: We rephrase these sentences and removed “through detailed land-sea comparison”.

Specific comment #7

Line 89: rephrase “in contrast to previous thought”.

Response: We have deleted this sentence and rephrased this issue (lines 93-97).

Specific comment #8

Sentence line 139: add reference to a figure?

Response: We now include a figure reference (line 155-156).

Specific comment #9

Sentence line 142-144: Add a link to the previous sentence... we therefore utilize; or As a result, we utilize; re-stating the age model info seems unnecessary here – this has already been described by this point.

Response: We have modified this sentence as suggested.

Specific comment #10

Line 103: unprecedented.

Response: Done.

Specific comment #11

In the introduction, it would be helpful to the reader to explicitly talk about the aeolian dust sources (i.e., where are they? Include the source area on the map in Fig. 1?), and explain why we think the transition in sedimentology from red clay to loess/paleosol layers occurs around 2.6 Ma (i.e., the state-of-the-art with respect to this). Also, what drives the switches between loess and paleosol layers post-2.6 Ma? Because we only see the sedimentary log on the depth scale, it’s hard to see whether these alternations are linked to G-IG cycles or not. All of this is perhaps very obvious to people who work on loess monsoon records all the time, but less so to people working for example in the marine realm or on paleobotanical records, so it would improve the accessibility and reach of the paper to explicitly mention this.

Response: We now discuss aeolian dust sources in the revised “Introduction” (lines 83-84). Source areas are also included in Fig. 1b. In the revised “Introduction”, we explain that magnetostratigraphic results from the Chinese Loess Plateau suggest a sedimentological transition from red clay to loess-paleosol layers at ~2.6 Ma (lines 80-83). We explain what drives alternations between loess and

palaeosol layers after 2.6 Ma (lines 104-107). We further show the Chongxin mean grain size record plotted against depth in Fig. 2a. We further revise Figure 2 by including L- and S-numbers in Fig. 2c on the age scale to refer to alternating loess and palaeosol layers after 2.6 Ma, respectively.

Specific comment #12

Fig. 1b: the red text on the map is difficult to read, I suggesting using another colour. Could dust source region be shown on the map?

Response: We revised Fig. 1, with inclusion of dust source regions on a topographic map (Fig. 1b).

Specific comment #13

Fig 2: I would consider using a more than three-point running average to smooth the data, so it's more comparable to the "global climate" curves plotted below. Also, see comments below about CO₂ and benthic d18O. Consider including the sedimentological info also on the age scale (like the paleomag).

Response: We now use a seven-point average to smooth the data in the revised Fig. 2c.

Specific comment #14

Fig. 3: consider adding horizontal lines for the main orbital periods on the wavelets. See comments below about CO₂, cross-wavelets, etc.

Response: We now use horizontal lines for the main orbital periods on the wavelets in Fig. 3.

Specific comment #15

Line 147 and more broadly; I would be careful referring to this curve as "the" reference atmospheric $p\text{CO}_2$ record. The $p\text{CO}_2$ curve of Berends et al 2021 is derived from an inverse forward modelling approach that deconvolutes the ice volume and deep-sea temperature signals in benthic foraminiferal $\delta^{18}\text{O}$ records. In the authors' words, this approach "*is a tool that determines how modelled CO₂ should have evolved over time in order to affect the global climate in such a way that the observed benthic $\delta^{18}\text{O}$ record is reproduced*". Whilst I appreciate that it is simpler to compare your record to and perform spectral analyses on a high-resolution, continuous synthetic $p\text{CO}_2$ curve, it seems important that you also compare your AWM record with data-based paleo- $p\text{CO}_2$ reconstructions, such as those compiled & homogenised by Sosdian et al., 2018 and Rae et al 2021 (possibly also including newer records for the Pliocene such as Guillermic et al 2022); see example figure below. I would strongly suggest including these $p\text{CO}_2$ data on Figure 2. The uncertainty on the modelled CO₂ curve (c.f. Fig 4 of Berends et al 2021), which is particularly high for the warm mid-Pliocene, should also be included in your figure. As is clear from Fig. 6 of Berends et al 2021, the different modelled CO₂ curves often show quite different trends from each other over iNHG which are not necessarily in agreement with proxy data, so it needs to be clear in your text and figures that the Berends CO₂ curve is a proposed scenario and NOT the record of past $p\text{CO}_2$ variations, as it is currently treated. For example, sentences/comments such as those lines 34, 147-149, 170, 241, 264, 279, etc need to be more nuanced with respect to this.

Response: We have changed this CO₂ issue as suggested (lines 222-254 and Fig. 2).

Specific comment #16

Arguably, rather than carrying out spectral analyses on this modelled CO₂ scenario derived from benthic d18O, it might be more direct and make more sense to compare the spectral characteristics of your grain size record with the (more robust) benthic d18O compilation, taken to represent “global climate” variations. In this context, given the high-resolution age model presented for your new record, I wonder if it might be a good idea to carry out cross-spectral analyses (or cross-wavelets) between, for example, mean grain size and benthic d18O or SL (rather than singular spectral analyses/wavelets of each record). This would allow you to assess changing coherence and phase relationships over time.

Response: We have changed the spectral analyses of Chongxin mean grain size, global average sea level, and atmospheric CO₂ concentration to cross-wavelet spectral analysis of Chongxin mean grain size and global average sea level as suggested (lines 178-184 and Fig. 3).

Specific comment #17

Given the discussion line 222 onwards, it also might be nice to also compare the record to NH high-latitude sea surface temperature for example (e.g. the Herbert et al. high-latitude NH SST stack (2010,2016), or the Lawrence et al 2009 ODP Site 982 SST record, or a different one I am not aware of), and also plot this in Fig. 2.

Response: We now include the North Atlantic Ocean SST record (Lawrence et al., 2009) in Fig. 2.

Specific comment #18

Line 153 – it is not clear what you are referring to as “the younger loess-paleosol sequence”. It looks to me like the increase in variability in the detrended record shown starts gradually around 2.8-2.9 Ma; whereas the switch to higher “mean state” mean grain size occurs almost exactly at the Gauss-

Matuyama boundary ~2.6 Ma, coincident with the sedimentology change. It might be helpful to add (to Fig. 2) a detrended record that shows orbital-scale (not just millennial) variability to support this statement; and also, the global benthic $\delta^{18}\text{O}$ compilation record to illustrate the switch to larger-amplitude G-IG cycles that you describe. Also, the change in baseline IG mean grain size after 2.6 Ma warrants a mention.

Response: We changed “the younger loess-paleosol sequence” to be more specific: “the Pleistocene loess-paleosol sequence”. We also include detrended and orbitally filtered records of the Chongxin mean grain size and global sea level records, which both show orbital-scale variability in the revised Fig. 2. We further modified related discussion, as suggested, including discussion of grain size shifts at 2.7 Ma and 2.6 Ma, and changes in baseline G-IG mean grain size after 2.6 Ma (lines 168-178).

Specific comment #19

Line 157: To what extent is the switch in mean grain size at ~2.6 Ma mechanistically linked to the switch in sedimentology (clay to loess-paleosol)? Are the two variables necessarily linked (i.e., are you saying that the switch in sedimentology is a direct and complete result of the change in winter monsoon wind strength and increased dust deposition)? As I mentioned for the introduction, this may seem obvious to CLP specialists but is not clear to me. Is the lithological change as abrupt as it appears on the log or is it more gradual, like the grain size change?

Response: We now further clarify the link between grain size and sedimentological shifts across the iNHG (lines 185-194).

Specific comment #20

Line 159: “and ASM weakening” – needs an explanation, not intuitive

Response: We change “ASM weakening” to “ASM weakening as indicated by distinct decreases in magnetic susceptibility (Sun et al., 2006), rates of chemical weathering (An et al., 2001; Wang et al., 2022), and soil carbonate $\delta^{13}\text{C}$ values (An et al., 2001)” (lines 191-192).

Specific comment #21

Line 164: “A thick higher”... is this a mistake?

Response: We remove this sentence to avoid repetition.

Specific comment #22

Line 173 – this is not clear on the wavelet figures, perhaps add horizontal lines denoting the orbital periods you discuss.

Response: We include horizontal lines to denote our discussed orbital periodicities (Fig. 3).

Specific comment #23

Line 180-181: this isn't really clear from the raw data figure (that the 100-kyr cycles are obliquity bundles) – maybe illustrate this somehow with a filter or annotation?

Response: We remove this statement.

Specific comment #24

Line 185: shouldn't this uncertainty in grain-size measurements be included on figure 2 (raw and filtered data)?

Response: We include grain-size measurement uncertainties in Supplementary Fig. 10.

Specific comment #25

One thing that I think would add to the paper would be a final figure that compares the new record with other monsoon records spanning iNHG. What to include is of course up to the authors (perhaps some of the ones mentioned in the discussion?), but I think it would help put the new record into a regional (land & ocean) context, in terms of how it advances the state of the art on monsoon evolution.

Response: See response to specific comment #5.

Specific comment #26

Line 243; add reference to new figure with paleo- $p\text{CO}_2$ data.

Response: These are now included.

Specific comment #27

Line 246: missing word?

Response: We have removed this statement from the revised manuscript.

Specific comment #28

Line 286: can you really say this with certainty? It seems quite hard to decouple to influence of global climate cooling and NH ice sheet growth from concurrent $p\text{CO}_2$ decreases.

Response: In our revision, we rephrased and toned down this argument (lines 222-254).

References cited in our response above

- An, Z.S., Kutzbach, J.E., Prell, W.L. and Porter, S.C., 2001. Evolution of Asian monsoons and phased uplift of the Himalaya-Tibetan plateau since Late Miocene times. *Nature*, 411: 62–66.
- Lawrence, K.T., Herbert, T.D., Brown, C.M., Raymo, M.E. and Haywood, A.M., 2009. High-amplitude variations in North Atlantic sea surface temperature during the early Pliocene warm period. *Paleoceanography*, 24(2): PA2218.
- Sun, Y.B., Clemens, S.C., An, Z.S. and Yu, Z.W., 2006. Astronomical timescale and palaeoclimatic implication of stacked 3.6-Myr monsoon records from the Chinese Loess Plateau. *Quaternary Science Reviews*, 25(1-2): 33–48.
- Wang, K.X., Lu, H.Y., Lei, F., Lyu, H.Z., Wang, H.L. and Wang, Y.C., 2022. East Asian monsoon precipitation decrease during Plio-Pleistocene transition revealed by changes in the chemical weathering intensity of Red Clay and loess-paleosol. *Palaeogeography Palaeoclimatology Palaeoecology*, 601: 111080.

Reviewer #3

Overview:

The East Asian winter and summer monsoon (EAWM/EASM) affects the lives of millions of people, so understanding how the EAWM/EASM operated in the past is important in a warming climate that now more and more resembles conditions at the end of the Pliocene in terms of both CO₂ levels and average temperatures. Ao and co-workers present a new, high resolution grain size record (representing EAWM) with a strong paleomagnetic age control from the central part of the Chinese Loess Plateau, spanning a period of 3.6-1.9 Ma (late Pliocene/early Pleistocene). They find that late Pliocene–early Pleistocene EAWM is characterized by persistent 41-kyr and ~100-kyr cycles, and superimposed on orbital variability, millennial timescale EAWM fluctuations occurred much earlier than previously thought. The data presented in the manuscript reflect the results of very significant field and laboratory work, with the authors performing paleomagnetic measurements of 251 samples and laser diffraction analysis of 3487 loess/paleosol/red clay samples, which makes this study impressive and unique. The text of the manuscript is very well written, clear and easy to follow, and most of the authors' conclusions are logical and scientifically sound. There are of course some issues that need to be clarified in a minor revision, but in my opinion (with perhaps one exception) they are not of such importance that they would fundamentally affect the main conclusions of the paper. My concerns are primarily related to chapter “Orbital-scale AWM dynamics” (lines 273-279) and Figure 2, and I think the text needs some minor amendments:

Response: We thank Reviewer #3 for positive feedback and comments about our manuscript. We have revised the manuscript as suggested.

Specific comment #1

1) From Figure 2c, it can be seen that interglacial intensities were similar before/after the iNHG or perhaps a little less intense after 2.7 Ma. As for the glacials, they clearly became more intense after 2.7 Ma. So, the statement that both glacials and interglacials became more intense (“marked stepwise increase in both glacial and interglacial AWM intensity”) is not evident in the MGS record, only that glacials became more intense and the amplitudes of fluctuations between glacials/interglacials. This must be addressed during the revisions.

Response: We agree that glacial MGS changes across the iNHG are substantially larger than interglacial MGS changes, similar to the sea level record (Fig. 2). We have changed the statement, as suggested, to “glacial AWM became more intense after the iNHG”.

Specific comment #2

2) Based on Fig. 2, the MGS record resembles the sea level curve, with larger amplitude fluctuations after 2.7 Ma. This is in marked contrast to the CO₂ record showing larger (sometimes >100 ppm) swings before the iNHG, while smaller scale fluctuations (~40 ppm) after 2.7 Ma. At the same time, CO₂ concentrations fluctuated around a lower mean value (~260 ppm) after the iNHG indicating significant changes in the GHG boundary conditions (likewise ice volume), leading to a different climate state. Is this what the authors are referring to?

Response: We now include the derived CO₂ curve from high-resolution benthic $\delta^{18}\text{O}$ records using the inverse forward modelling approach (Berends et al., 2021), as well as data-based CO₂ reconstructions (Bartoli et al., 2011; de la Vega et al., 2020; Dyez et al., 2018; Guillermic et al., 2022; Martínez-Botí et al., 2015; Sosdian et al., 2018) in the revised Fig. 2. Larger-amplitude orbital oscillations before 2.7 Ma in the CO₂ record of Berends et al. (2021) derived from benthic $\delta^{18}\text{O}$ records

is not observed in the benthic $\delta^{18}\text{O}$ and global mean sea level records, nor in the data-based CO_2 reconstructions, so it may be a modelling artefact. Therefore, we now only compare the Chongxin MGS record with global mean sea level record and not with CO_2 variability on the orbitally oscillating amplitudes. However, we do use sea level and both proxy- and model-based CO_2 reconstructions to refer to different climate-cryosphere boundary conditions before and after the iNHG, because it has been well documented that Northern Hemisphere glaciation was enhanced when atmospheric CO_2 concentrations decreased across the iNHG. We find that this climate-cryosphere boundary condition change did not alter orbital-scale AWM periodicities but substantially enhanced glacial AWM intensity across the iNHG.

Specific comment #3

Other, less important questions also arise in relation to the interpretations or methodology: 3) This is just a pedantic point, but the authors describe in several places (e.g. lines 68, 83) that the record they examine is continuous. Unfortunately, this is only an assumption. It would be difficult to imagine a dynamic eolian environment such as the CLP without shorter/longer (up to at least decades or a few hundred years) erosional periods over the last 3.6 million years. There is direct evidence for sedimentation gaps at the CLP margin based on luminescence dating results (see Stevens et al., 2018 and references therein). Therefore, I suggest using the term “quasi-continuous” in the text.

Response: We have removed “continuous” from or changed it to “quasi-continuous”, with a mention of potential erosional hiatuses in sections on the northern desert margin of the CLP (Stevens et al., 2018) (lines 101-104).

Specific comment #4

4) The authors state that “Mean grain sizes of bulk samples and extracted quartz particles have almost identical variation for both red clay and loess-palaeosol sequences on the CLP”, by referring to Sun et al. (2010). In my view, this should have been tested using a subset of their samples to prove that this is true at the study site. For instance, bulk and quartz grain size variations were shown to be different in European loess (Újvári et al., 2016, 2017). I'm not stating that this holds true for the CLP, but it would have been worth testing.

Response: Mean grain size variations of quartz particles and bulk samples from different sections across the CLP are largely comparable (e.g., Ding et al., 2002; Sun et al., 2006, 2010), such as the Xifeng and Lingtai sections in Fig. 1 below. This is different from European loess. Quartz particle extraction from thousands of bulk samples is labour intensive work that is not feasible for the Chongxin section, so we use bulk sample grain sizes to infer AWM variability.

Fig. 1. Mean grain size variations of quartz particles (black line) and bulk samples (blue line) from the Xifeng and Lingtai loess-paleosol/red clay sections (Sun et al., 2006).

Specific comment #5

5) Line 151-153, “coarser grain size-stronger winds relation”: I agree that stronger wind is one of the most likely explanations, in addition to other factors that influence the transport capacity of the medium, including, for example, the state of the eolian surface and temperature of the transport medium. At lower temperatures an aeolian surface is more active, cohesive forces are weaker and the air is denser, so under colder conditions larger particles can be transported (see 3.1.1.4 subchapter in Ujvari et al., 2016).

Response: We have changed the wording to “indicative of increased dust transport capacity by stronger AWM winds from drier source regions located to the north and west to the CLP under colder conditions across the iNHG (Ding et al., 1995; Hao et al., 2012; Sun et al., 2010; Ujvári et al., 2016; Xiao et al., 1995)” (lines 170-172).

Specific comment #6

6) Methods section, grain size analyses (lines 305-309): Provide information on instrument parameters (absorption and refractive index). Have you used the Mie theory (or Fraunhofer approximation) for GSD calculations?

Response: In the revised Methods section, we now include details of grain size analyses, including related instrument parameters and a mention of the Mie theory used for GSD calculations (lines 321-332).

Specific comment #7

7) Methods section, age model (lines 377-379): Orbital tuning may have resulted in increased power at the 41 ka band, nicely seen in Fig S9. Could you comment on this?

Response: We comment on this in the revised Method section (lines 425-429).

Specific comment #8

8) Methods section, 8 ka lag (lines 381-382): This lag is debated, others argue for a shorter lag (~5 ka) based on absolute ages (Stevens et al., 2018). However, the ~ 5 ka lag was proposed for the response of EASM to insolation forcing.

Response: We agree that the lag between orbital forcing and CLP grain size (AWM) remains debated, particularly before 1 Ma. Following previous orbital tunings of CLP loess-palaeosol grain size records with the same procedure (Ding et al., 2002; Sun et al., 2006), we also adopt the 8-kyr-lagged obliquity curve, which refers to the SPECMAP-defined lag for obliquity (Imbrie et al., 1984), for tuning, like previous studies. However, obliquity tuning using 8-kyr- or 5-kyr-lagged obliquity results in only very slight differences (~3 kyr) in the related astronomical age models, which does not influence our interpretations. Nevertheless, given that detailed luminescence dating results suggest that EASM lags precession-paced summer insolation by ~5 kyr (Stevens et al., 2018), which means that the EASM lags short-duration precession by ~5 kyr, and likely thus longer-duration obliquity by >5 kyr, the commonly adopted 8 kyr lag between Asian monsoon and obliquity appears to enable more reasonable tuning results.

Specific comment #9

9) Methods, age model (lines 386): What was the accepted tolerance of paleomagnetic boundary shifts in kyrs?

Response: We now specify that our tolerance of paleomagnetic boundary shifts caused by tuning is less than two obliquity cycles (<80 kyr) (lines 412-413). However, our orbital tuning generally shifted the ages of major palaeomagnetic reversal boundaries (polarity chrons or subchrons) by <30 kyr (Supplementary Table 1). The age of the short-duration Réunion geomagnetic excursion is ~60 kyr younger in our astronomical timescale than in the 2020 GPTS (Gradstein et al., 2020). This slightly larger shift of the Réunion geomagnetic excursion may be due to increased age uncertainties of geomagnetic excursions (Laj and Channell, 2007; Roberts, 2008) and relatively more variable positions and post-depositional NRM lock-in depths in CLP aeolian sediments (Jin et al., 2019; Pan et al., 2021).

Specific comment #10

Technical corrections, typos

line 103: “unprecedented” should be replaced by “unprecedented”

line 137: write “AMW indicator”

line 192: consider writing “2.66 Ma” instead of “2.7 Ma”

line 245: “CO₂ concentration increase”, do you mean “decrease”?

line 246: write “ice volume and atmospheric CO₂”

line 374: write “which were assigned”

line 641: replace “Loss” by “Loess”

Response: We have included these helpful edits in the new version of the text.

Specific comment #11

lines 259-260: consider using “different control” instead of “larger control”

Response: On rereading this sentence, we found that this argument did not make that much sense, so we have deleted it.

Specific comment #12

Figure 1: I suggest changing the label of Fig. 1b to make sure these are readable. What does the hatching mean?

Response: We include a revised Fig. 1, which no longer contains hatching.

References cited in our response above

- Bartoli, G., Hönisch, B. and Zeebe, R.E., 2011. Atmospheric CO₂ decline during the Pliocene intensification of Northern Hemisphere glaciations. *Paleoceanography*, 26(4): PA4213.
- Berends, C.J., de Boer, B. and van de Wal, R.S.W., 2021. Reconstructing the evolution of ice sheets, sea level, and atmospheric CO₂ during the past 3.6 million years. *Climate of the Past*, 17(1): 361–377.
- de la Vega, E., Chalk, T.B., Wilson, P.A., Bysani, R.P. and Foster, G.L., 2020. Atmospheric CO₂ during the Mid-Piacenzian Warm Period and the M2 glaciation. *Scientific Reports*, 10(1): 11002.
- Ding, Z.L., Derbyshire, E., Yang, S.L., Yu, Z.W., Xiong, S.F. and Liu, T.S., 2002. Stacked 2.6-Ma grain size record from the Chinese loess based on five sections and correlation with the deep-sea $\delta^{18}\text{O}$ record. *Paleoceanography*, 17(3): doi:10.1029/2001PA000725.
- Ding, Z.L., Liu, T.S., Rutter, N.W., Yu, Z.W., Guo, Z.T. and Zhu, R.X., 1995. Ice-volume forcing of East Asian winter monsoon variations in the past 800,000 Years. *Quaternary Research*, 44(2): 149–159.
- Dyez, K.A., Hönisch, B. and Schmidt, G.A., 2018. Early Pleistocene obliquity-scale pCO₂ variability at~ 1.5 million years ago. *Paleoceanography and Paleoclimatology*, 33(11): 1270–1291.
- Gradstein, F.M., Ogg, J.G., Schmitz, M.D. and Ogg, G.M., 2020. *Geologic Time Scale 2020*. Elsevier.
- Guillermic, M., Misra, S., Eagle, R. and Tripathi, A., 2022. Atmospheric CO₂ estimates for the Miocene to Pleistocene based on foraminiferal $\delta^{11}\text{B}$ at Ocean Drilling Program Sites 806 and 807 in the Western Equatorial Pacific. *Climate of the Past*, 18(2): 183–207.
- Hao, Q.Z., Wang, L., Oldfield, F., Peng, S.Z., Qin, L., Song, Y., Xu, B., Qiao, Y.S., Bloemendal, J. and Guo, Z.T., 2012. Delayed build-up of Arctic ice sheets during 400,000-year minima in insolation variability. *Nature*, 490(7420): 393–396.
- Jin, C.S., Liu, Q.S., Xu, D.K., Sun, J.M., Li, C.G., Zhang, Y., Han, P. and Liang, W.T., 2019. A new correlation between Chinese loess and deep-sea $\delta^{18}\text{O}$ records since the middle Pleistocene. *Earth and Planetary Science Letters*, 506: 441–454.
- Laj, C. and Channell, J.E.T., 2007. Geomagnetic excursions. In: *Treatise in Geophysics: Volume 5, Geomagnetism* (editor: M. Kono). Chapter 10, Elsevier, Amsterdam: 373–416.
- Martínez-Botí, M.A., Foster, G.L., Chalk, T.B., Rohling, E.J., Sexton, P.F., Lunt, D.J., Pancost, R.D., Badger, M.P.S. and Schmidt, D.N., 2015. Plio-Pleistocene climate sensitivity evaluated using high-resolution CO₂ records. *Nature*, 518(7537): 49–54.
- Pan, Q., Xiao, G.Q., Zhao, Q.Y., Chen, R.S., Ao, H., Shen, Y.F., Cheng, J.Y. and Zhu, Z.M., 2021. The Jaramillo subchron in Chinese loess-paleosol sequences. *Palaeogeography Palaeoclimatology Palaeoecology*, 572: 110423.
- Roberts, A.P., 2008. Geomagnetic excursions: knowns and unknowns. *Geophysical Research Letters*, 35: L17307.

- Sosdian, S.M., Greenop, R., Hain, M.P., Foster, G.L., Pearson, P.N. and Lear, C.H., 2018. Constraining the evolution of Neogene ocean carbonate chemistry using the boron isotope pH proxy. *Earth and Planetary Science Letters*, 498: 362–376.
- Sun, Y.B., An, Z.S., Clemens, S.C., Bloemendal, J. and Vandenberghe, J., 2010. Seven million years of wind and precipitation variability on the Chinese Loess Plateau. *Earth and Planetary Science Letters*, 297(3-4): 525–535.
- Sun, Y.B., Clemens, S.C., An, Z.S. and Yu, Z.W., 2006. Astronomical timescale and palaeoclimatic implication of stacked 3.6-Myr monsoon records from the Chinese Loess Plateau. *Quaternary Science Reviews*, 25(1-2): 33–48.
- Ujvári, G., Kok, J.F., Varga, G. and Kovács, J., 2016. The physics of wind-blown loess: Implications for grain size proxy interpretations in Quaternary paleoclimate studies. *Earth-Science Reviews*, 154: 247–278.
- Xiao, J.L., Porter, S.C., An, Z.S., Kumai, H. and Yoshikawa, S., 1995. Grain size of quartz as an indicator of winter monsoon strength on the loess plateau of central China during the last 130000 yr. *Quaternary Research*, 43: 22–29.

REVIEWERS' COMMENTS

Reviewer #2 (Remarks to the Author):

The authors have done a very thorough job revising their manuscript in line with all of the reviewers' comments, and the revised paper is clearer, more accessible, and more impactful. I therefore recommend it for publication in its present form, with a few small comments below.

Revised Figures 1b and 1c are a great addition, and are much more informative with respect to dust sources. To make this figure even better, I suggest putting a black outline box on Map a showing the area in Map b, and in Map b denoting the area blown up in Map c!

Revised Figure 2 is also much improved in my view, and my comments have been taken on board. The new pCO₂ figure is very nice and much more representative of actual proxy reconstructions. I suggest putting more intuitive axis titles for panels d and f showing the filtered records, instead of "fitting" (e.g., xx-xx kyr filtered MGS). Is the text that says "Red clay" below record c supposed to be there? If so, I wasn't sure why. The inclusion of the 982 SST record is nice, please check that you used the updated age model in Herbert 2016.

I don't think that "northern hemisphere" should be systematically capitalised. Same for late Pliocene and early Pleistocene, as they are not official sub-epochs.

Line 176 "stronger inferred AWM winds"?

The cross-wavelet analysis between MGS and sea level is a nice addition to the paper.

The new Supp. Fig 11 comparing different iNHG monsoon records is really interesting and definitely opens the floor for discussion. Nice addition.

Supp. Fig. 8 – I don't quite understand why the grey dashed lines, which presumably show tie points (not specified in the caption), don't match up with the red dots ("tuning points")

Clara Bolton

Reviewer #3 (Remarks to the Author):

The authors have duly addressed the issues raised in my review. I have no further comments.

We thank the reviewers and the editor for evaluating our manuscript. Please find below our point-by-point responses in blue. Our revisions in the main text are also marked in blue.

Reviewer #2

General comments:

The authors have done a very thorough job revising their manuscript in line with all of the reviewers' comments, and the revised paper is clearer, more accessible, and more impactful. I therefore recommend it for publication in its present form, with a few small comments below.

Response: We thank Reviewer #2 for their positive evaluation and helpful comments.

Specific comment #1

Revised Figures 1b and 1c are a great addition, and are much more informative with respect to dust sources. To make this figure even better, I suggest putting a black outline box on Map a showing the area in Map b, and in Map b denoting the area blown up in Map c!

Response: We have further revised Fig. 1 as suggested.

Specific comment #2

Revised Figure 2 is also much improved in my view, and my comments have been taken on board. The new pCO₂ figure is very nice and much more representative of actual proxy reconstructions. I suggest putting more intuitive axis titles for panels d and f showing the filtered records, instead of "fitting" (e.g., xx-xx kyr filtered MGS). Is the text that says "Red clay" below record c supposed to be there? If so, I wasn't sure why. The inclusion of the 982 SST record is nice, please check that you used the updated age model in Herbert 2016.

Response: We have revised panels d and f as suggested. "Red clay" below record c has been removed. We now adopt the updated age model of the 982 SST record by Herbert et al. (2016).

Specific comment #3

I don't think that "northern hemisphere" should be systematically capitalised. Same for late Pliocene and early Pleistocene, as they are not official sub-epochs.

Response: We have changed "Northern Hemisphere", "Late Pliocene", and "Early Pleistocene" to "northern hemisphere", "late Pliocene", and "early Pleistocene", respectively, throughout.

Specific comment #4

Line 176 "stronger inferred AWM winds"?

Response: This indicates more significant AWM intensification during glacials compared to interglacials across the iNHG. We have included this information in the revised manuscript (lines 164-165).

Specific comment #5

The cross-wavelet analysis between MGS and sea level is a nice addition to the paper. The new Supp. Fig 11 comparing different iNHG monsoon records is really interesting and definitely opens the floor for discussion. Nice addition.

Response: We thank Reviewer #2 for their positive evaluation on these revisions.

Specific comment #6

Supp. Fig. 8 – I don't quite understand why the grey dashed lines, which presumably show tie points (not specified in the caption), don't match up with the red dots ("tuning points").

Response: We have changed the now green (previous grey) dashed lines to match the red dots (Supplementary Fig. 8).